# Process-based analysis of terrestrial carbon flux predictability

István Dunkl[1,2], Aaron Spring[1], Pierre Friedlingstein[3], and Victor Brovkin[1,4]

[1]Max Planck Institute for Meteorology, Hamburg, Germany
[2]International Max Planck Research School on Earth System Modelling, Hamburg, Germany
[3]College of Engineering, Mathematics and Physical Sciences, University of Exeter, Exeter, UK
[4]Center for Earth System Research and Sustainability, University of Hamburg, Germany

**Correspondence:** István Dunkl (istvan.dunkl@mpimet.mpg.de)

**Abstract.** Despite efforts to decrease the discrepancy between simulated and observed terrestrial carbon fluxes, the uncertainty in trends and patterns of the land carbon fluxes remains high. This difficulty raises the question to what extent the terrestrial carbon cycle is predictable, and which processes explain the predictability. Here, the perfect model approach is used to assess the potential predictability of net primary production (NPP*pred*) and heterotrophic respiration (Rh*pred*) by using ensemble

simulations conducted with the Max-Planck-Institute Earth System Model. In order to assess the role of local carbon flux predictability (CF*pred*) on the predictability of the global carbon cycle, we suggest a new predictability metric weighted by the amplitude of the flux anomalies. Regression analysis is used to determine the contribution of the predictability of different environmental drivers to NPP*pred* and Rh*pred* (soil moisture, air temperature and radiation for NPP and soil organic carbon, air temperature and precipitation for Rh). Global NPP*pred* is driven to 62 and 30% by the predictability of soil moisture and

temperature, respectively. Global Rh*pred* is driven to 52 and 27% by the predictability of soil organic carbon and temperature, respectively. The decomposition of predictability shows that the relatively high Rh*pred* compared to NPP*pred* is due to the generally high predictability of soil organic carbon. The seasonality in NPP*pred* and Rh*pred* patterns can be explained by the change in limiting factors over the wet and dry months. Consequently, CF*pred* is controlled by the predictability of the currently limiting environmental factor. Differences in CF*pred* between ensemble simulations can be attributed to the occurrence of wet

and dry years, which influences the predictability of soil moisture and temperature. This variability of predictability is caused by the state dependency of ecosystem processes. Our results reveal the crucial regions and ecosystem processes to be considered when initializing a carbon prediction system.

## 1   Introduction

As a net sink for atmospheric $CO_2$, terrestrial ecosystems absorb around one third of the anthropogenic emissions (Friedling-

stein et al., 2020). Carbon fluxes between the land-atmosphere interface have a high interannual variability with a standard deviation (SD) of 0.7 PgC yr$^{-1}$ (Sitch et al., 2015) and cause the majority of the atmospheric $CO_2$ fluctuations (Ciais et al., 2013; Spring et al., 2020). The high variability of terrestrial carbon fluxes can be attributed to the sensitivity of land surface processes to climatic drivers, however the relative importance of temperature and precipitation are still debated (Jones et al., 2001; Beer et al., 2010; Bloom et al., 2016; Fang et al., 2017; Jung et al., 2017; Bastos et al., 2018). In accordance with the

limited understanding of carbon flux variability, models are not able to fully reproduce the spatiotemporal patterns of the terrestrial carbon cycle. This is reflected in the poor representation of soil organic carbon (SOC) in Earth System Models (ESM), the inability to adequately model gross primary production (GPP) from eddy covariance flux tower sites (Luo et al., 2015) and the difficulty to detect the efforts taken in emission reduction due to internal variability of atmospheric $CO_2$ variability (Spring et al., 2020). In order to produce more realistic predictions, efforts in model development have been directed towards using

observations to constrain model parameters (Zeng et al., 2014; Bloom et al., 2016; Mystakidis et al., 2016; Chadburn et al., 2017; Tziolas et al., 2020), and to refine model structure to incorporating more processes and interactions (Krull et al., 2003; Stockmann et al., 2013; Xu et al., 2014; Luo et al., 2016). While efforts in model development are continuing to narrow the gap between the simulated and observed carbon cycle, the lack of progress in improving the predictive ability of the models raises the question to what extent the terrestrial carbon cycle is predictable at all (Luo et al., 2015).

The potential predictability of a system can be estimated by using the perfect model framework. Ensemble simulations are initialized along a control run with each member of the ensemble having slightly perturbed initial conditions. The upper limits of predictability are then derived by analysing the divergence of the ensemble simulations. This method assumes (a) perfect model physics which are able to reproduce the full spectrum of natural variability and (b) perfect knowledge of the modelled system and a model whose representation of the real world is "perfect enough" (Boer et al., 2013). Séférian et al.

(2018) used the perfect model framework to assess the potential predictability of terrestrial carbon fluxes (CF*pred*) at annual time-steps. They estimated the predictive horizon of terrestrial carbon fluxes to be two years globally and up to three years in northern latitudes. The high variability of predictability among different initializations suggests a state-dependence of CF*pred*, but no further mechanisms of predictability were investigated therein. Multiple processes can being considered as the sources of CF*pred*. Due to the high sensitivity of the terrestrial carbon cycle to climate, climate predictability provides carbon fluxes

with a basic prediction horizon. The main contributor to climate predictability is El Niño-Southern Oscillation (ENSO), which explains over 40% of the variability in global net primary production (NPP) (Bastos et al., 2013) and a large fraction of CF*pred* (Zeng et al., 2008). El Niño events are associated with high temperatures and low precipitation in the tropics which cause a reduction of the land carbon sink of 1.8 PgC $yr^{-1}$ per 1 °C sea surface temperature (SST) anomaly in the Niño 3 region (Jones et al., 2001). This strong relationship between SST and the carbon cycle was used by Betts et al. (2016) to predict annual $CO_2$

growth. Their statistical model uses the annual average SST in the Niño 3.4 region to successfully predict the $CO_2$ rise with a precision of 0.53 ppm $yr^{-1}$. Furthermore, Spring and Ilyina (2020) showed that ESM-based initialized predictions can predict atmospheric $CO_2$ variations up to three years in advance.

   However, CF*pred* is extended beyond the predictability of climate by slowly varying land surface processes that filter out the high-frequency noise of the climate signal. As the most prominent process, soil moisture memory is known to increase the

predictability of temperature (TEMP*pred*) and precipitation (PRECIP*pred*) by several months (Chikamoto et al., 2015), but memory can also be attributed to phenology (Weiss et al., 2014) and SOC (Lovenduski et al., 2019). Besides the slowly changing land state variables, the memory is further extended through land-atmosphere coupling which propagates soil anomalies back to to the atmosphere by energy and water fluxes (Bellucci et al., 2015).

Previous studies that focus on the mechanisms of CF*pred* investigated the role of various land processes and how they contribute to the overall CF*pred*. Weiss et al. (2014) found increased predictability of evaporation and to some extent temperature due to a dynamic simulation of leaf area index (LAI), which would also extend CF*pred*. The role of land surface initialization on CF*pred* was studied by Zeng et al. (2008) and Lovenduski et al. (2019). Zeng et al. (2008) isolated the fraction of CF*pred* which is based solely on initial conditions and compare fully coupled dynamic simulations with statistical models. Lovenduski et al. (2019) quantified the degree to which CF*pred* improves when the land surface is initialized. They also assessed the relative importance of the individual land surface processes for the variability of terrestrial carbon fluxes and found that CF*pred* depends on the correct initialization of vegetation carbon biomass and soil moisture rather than temperature. These studies have shown the significant advantage of dynamic forecasting systems, suggesting CF*pred* extends beyond the predictability of the forcing variables due to land surface processes. However, these studies were not focused on contributions of individual drivers of carbon fluxes to CF*pred* or on processes responsible for maintaining CF*pred*.

Here, we use perfect model simulations conducted with an ESM to investigate the structure and mechanisms of the CF*pred*. Initialized ensemble simulations are created from a range of ENSO states. Analysed are the carbon fluxes with the highest contribution to the interannual variability of the land-atmosphere $CO_2$ exchange. These are NPP with an interannual SD of 0.99 $PgCyr^{-1}$ and heterotrophic respiration (Rh) with an SD of 0.29 $PgCyr^{-1}$ (Wang et al., 2016). The potential predictability of NPP (NPP*pred*) and Rh (Rh*pred*) is derived from rate of divergence within the ensemble members. We evaluate the predictability data to find how NPP*pred* and Rh*pred* differ in their spatiotemporal patterns and variability. Lastly, we identify the key drivers of NPP and Rh and determine their contribution to NPP*pred* and Rh*pred*. We use this framework to explain the attained spatiotemporal patterns of CF*pred* and identify the underlying land system processes producing these patterns.

## 2    Methods

### 2.1    Earth system model

This study is based on the output of the MPI-ESM version 1.2 developed for the Coupled Model Intercomparison Project 6 (Mauritsen et al., 2019). The model runs fully coupled in the LR configuration that uses the atmospheric component ECHAM 6.3.05 with a T63 spatial truncation and 47 atmospheric layers. The atmospheric model is directly coupled with the land model JSBACH 3.20 and uses an interactive carbon cycle, which means atmospheric $CO_2$ reacts to land and ocean carbon fluxes.

### 2.2    Predictability metrics

The control simulation used in this study is a 1000–year unforced simulation with a preindustrial $CO_2$ concentration of 285 ppm. A total of 35 10-member ensemble simulations are initialized, each starting in January with a run time of two years. The unperturbed simulation of the control run is added to the ensembles as the 11th member. Initialization dates are selected manually in order to attain a diversity of ENSO states. The selected dates are grouped into three categories: El Niño, La Niña or ENSO-neutral.

The potential predictability is assessed by using a correlation-based and a distance-based metric. The Anomaly Correlation Coefficient (ACC) is a commonly used metric to measure forecast skill (Jolliffe and Stephenson, 2012) which calculates the correlation between predicted and observed anomalies as

$$ACC_{j,t} = \frac{cov(f,o)}{\sigma_f \cdot \sigma_o},\tag{1}$$

where $j$ and $t$ are grid cell and lead time, $cov$ is the covariance and $f$ and $o$ the forecast and validation anomalies. Similar to (Collins and Sinha, 2003; Becker et al., 2013), the noise in the ACC is reduced by averaging over several ACC values. This is achieved by taking all eleven ensemble members as the validation in turn, while the mean of the remaining ensemble members serves as the forecast. Although the ACC is an intuitive metric which is calculated from all initializations and thus provides a robust estimation of the predictability, it does not allow to investigate the variability of predictability between initializations. The comparison of predictabilities between initialization is achieved by the use of a distance based metric which is computed for all initializations individually. The distance based metric used here is the normalized ensemble variance $(V(t))$ based on the method proposed by Griffies and Bryan (1997). Predictability is defined as the ensemble variance normalized by the variance of the climatology as

$$V(t) = \frac{\frac{1}{M}\sum_{i=1}^{M}\left[X_i(t) - \overline{X}(t)\right]^2}{\sigma^2},\tag{2}$$

where $t$ is lead time, $M$ the number of ensemble members, $X_i$ the $i$th member, $\overline{X}$ the ensemble mean and $\sigma^2$ the variance of the control simulation. In this study, the complement of the normalized ensemble variance is used as: $V_c(t) = 1 - V(t)$. The resulting metric indicates perfect predictability at a value of 1, and an ensemble spread that exceeds the climatological variance for values below zero.

While ACC and $V_c$ allow the estimation of regional predictability, these metrics are not suitable to evaluate the impact of local predictabilities on the predictability of the global carbon cycle. This is due to the disregard of the flux amplitude in the calculation of the metrics. Both of the metrics are prone to producing above average predictabilities in regions where carbon fluxes are generally low or even close to zero, such as subtropical deserts. Here we propose a weighted predictability metric that allows to assess local predictabilities with regard of their impact on the predictability of the global carbon cycle. $V_c$ is weighted by using an approach similar to risk assessment, which is calculated as the product of likelihood and impact. Here a weighted predictability $wV_c$ is calculated by multiplying $V_c$ with the absolute carbon flux anomaly of the ensemble mean:

$$wV_c(t) = V_c(t) \times |\Delta FLUX(t)|.\tag{3}$$

## 2.3 Decomposition of predictability

In order to investigate the drivers of CF*pred*, the $V_c$ of NPP and Rh are decomposed into components contributing to the predictability of these fluxes similar to an approach used by Jung et al. (2017). They used regression analysis to determine the contribution of environmental variables to the anomalies in GPP and ecosystem respiration. Here the assumptions of Jung et al. (2017) are extended from carbon flux anomalies to CF*pred*: a high CF*pred* needs to be caused by a high predictability of

one or more of its driving environmental variables. Using this assumption, NPP*pred* and Rh*pred* are modeled as the response to the predictability of the individual environmental drivers. Regression analysis is used to determine the contribution of the predictability of the environmental variables to NPP*pred* and Rh*pred*. The drivers of NPP*pred* are selected following the drivers of GPP in Jung et al. (2017) as two layers of soil moisture (midSOIL*pred* for 19 – 78 cm depth and deepSOIL*pred* for 79 – 268 cm depth), air temperature (TEMP*pred*) and photosynthetically active radiation (PAR*pred*). The drivers of Rh*pred* are based on the rate modifying factors used in JSBACH to calculate Rh, which are TEMP*pred*, PRECIP*pred* and SOC*pred*. Although precipitation has no direct relationship with Rh, the Rh submodel used in JSBACH is parameterized using precipitation, because of its strong relationship with moisture in the uppermost soil layer where most of the respiration takes place. Instead of SOC, the content of the aboveground acid hydrolyzable carbon pool (here referred to as SOC) is used as a surrogate variable. The contribution of of the predictability of the environmental drivers to the CF*pred* is calculated as

$$V_cFLUX_{j,t,i} = \sum_k \left[ a_{j,t}^{DRI_k} \times V_cDRI_{k,j,t,i} \right] + \epsilon_{j,t,i} \tag{4}$$

with $V_cFLUX$ being the complementary normalized ensemble variance of NPP or Rh, $a^{DRI_k}$ coefficient of the $k$th driver (for example TEMP*pred*), $V_cDRI$ the predictability of the $k$th driver and $\epsilon$ the residual error term. Grid cell, lead time and initialization are denoted by the indices $j$, $t$ and $i$. The regression coefficients are calculated by using non-negative least squares (Mullen and van Stokkum, 2012) for every grid cell and lead time by using the data from all initializations. After fitting the regression model to the data, the individual components of CF*pred* are calculated as

$$V_cFLUX_{i,t,s}^{DRI_k} = a_{i,t}^{DRI_k} \times V_cDRI_{k,i,t,s}, \tag{5}$$

where $V_cFLUX^{DRI}$ describes the amount of predictability of $FLUX$ that can be contributed to the driver $k$.

## 3  Results and discussion

Out of the 35 ensemble simulations initialized along the control run, 7 simulations are part of the El Niño and 8 simulations part of the La Niña group (Fig. 1). The El Niño simulations peak between the September before initialization and January with peak values between 2.2 and 3.6 °C (three month running mean Niño 3.4 SST anomaly). They show a fast decline of the anomaly with most models having a negative anomaly in December of the first year and evolving into a La Niña event in the second year. Peaks of the La Niña simulations fall between September and June and, while their relative peak anomalies are smaller (-1.6 to 3.0 °C), the negative anomaly can be sustained well into the second year.

### 3.1  Potential predictability

The 35 perfect model simulations are used to assess potential NPP*pred* and Rh*pred*. Zonal means of the ACC are shown in Figure 2 (Zonal plots of predictability are limited to 30 ° South to 30 ° North to highlight the areas of high predictability). NPP*pred* and Rh*pred* are highest in the tropics between 20° North and South, where carbon fluxes are at their global maximum. However, apart from the generally high predictability in the tropics, the patterns of NPP*pred* and Rh*pred* differ in several

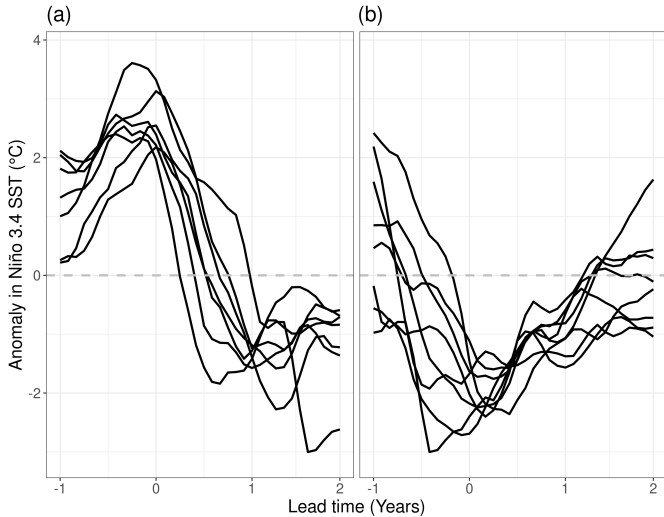

**Figure 1.** Three month running mean SST anomaly in the Niño 3.4 region of (a) 7 El Niño and (b) 8 La Niña simulations. Simulations are initialized at lead time 0 and run for 24 months. Lines show the Niño 3.4 SST of the control simulation.

aspects. While the ACC of NPP has a slower temporal decline with values above 0.8 for 2 to 3 months around the equator, the ACC of Rh drops below 0.5 within the first two months for most latitudes. However, Rh shows much higher long term predictability, especially in the second year of the simulation where Rh*pred* is much higher than NPP*pred*.

While both predictability patterns show signs of a seasonal cycle, they are out of phase with Rh*pred* distinctly following the

wet season and NPP*pred* appearing to be higher in the dry seasons of the first year. This has a large role on the comparability of NPP*pred* and Rh*pred*, since high NPP*pred* occurs at the time of the seasonal low of NPP fluxes, while high Rh*pred* is associated with the seasonal high. Another characteristic of the seasonal cycles is their continuity. Rh*pred* migrates continuously across the zones, while NPP*pred* demonstrates a sporadic behaviour with a high predictability at around 15° North in JFM and another one at 10° South in JAS.

The spatial patterns of ACC are shown in Figure 3 for March, June and September of the first year and September of the second year. Rh*pred* shows a very coherent pattern with a band of high predictability migrating from South to North across all continents. The patterns of NPP*pred* appear to be less constrained by latitude. Although March predictability is dominated by the northern tropics and subtropics, there are other high predictability regions based on initial memory, especially in high latitudes. As opposed to Rh*pred*, there is no high-predictability band moving across the zones. Instead, NPP*pred* is reemerging

south of the equator in September in the southern Amazon basin, southern Africa and Southeast Asia. An aberration from the seasonal pattern is in the Sahel which has a relatively high NPP*pred* throughout both years, except in June and July (not shown).

A large portion of the high NPP*pred* areas can be contributed to predictability gained by ENSO. These high predictability areas are concurring with the carbon flux anomalies caused by ENSO related climate variability (Hashimoto et al., 2004; Bastos

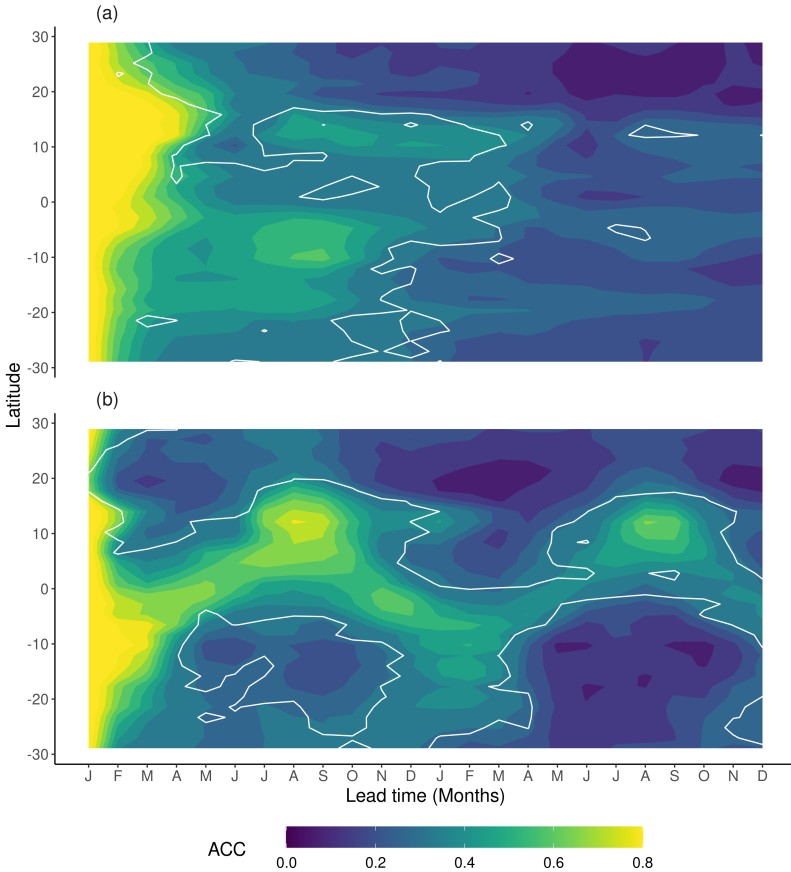

**Figure 2.** Zonal means of ACC derived from 35 ensemble simulations starting in January for (a) NPP and (b) Rh. Contour lines indicate correlations above the 95% confidence level.

et al., 2013). A specific example of this is the disparity in NPP*pred* between the tropical rainforests of the Amazon and the Congo basins. It shows that the high NPP*pred* of the tropics is not an intrinsic property of these ecosystems. A reason for the relatively low NPP*pred* within the Congo basin could be because it is not strongly impacted by ENSO (Holmgren et al., 2001). These findings highlight the importance of correctly simulating the ESNO process. Especially the localization of ENSO related rainfall patterns is crucial, since they provide a sustained and predictable anomaly in water availability.

Many of the identified spatial patterns of CF*pred* can be discovered in similar studies. Most models agree on the Amazon basin as the global hotspot of CF*pred* (Zeng et al., 2008; Ilyina et al., 2021), and some reflect the increased predictability in Southeast Asia and southern Africa (Zeng et al., 2008), but the comparison of predictability horizons remain difficult due to the use of different predictability metrics.

The results reveal different areas in which an operational NPP forecast can be used to increase food security. The high NPP*pred* of the Sahel and Kalahari savanna ecosystems (Fig. 3) could be used to plan stocking rates in order to avoid grassland

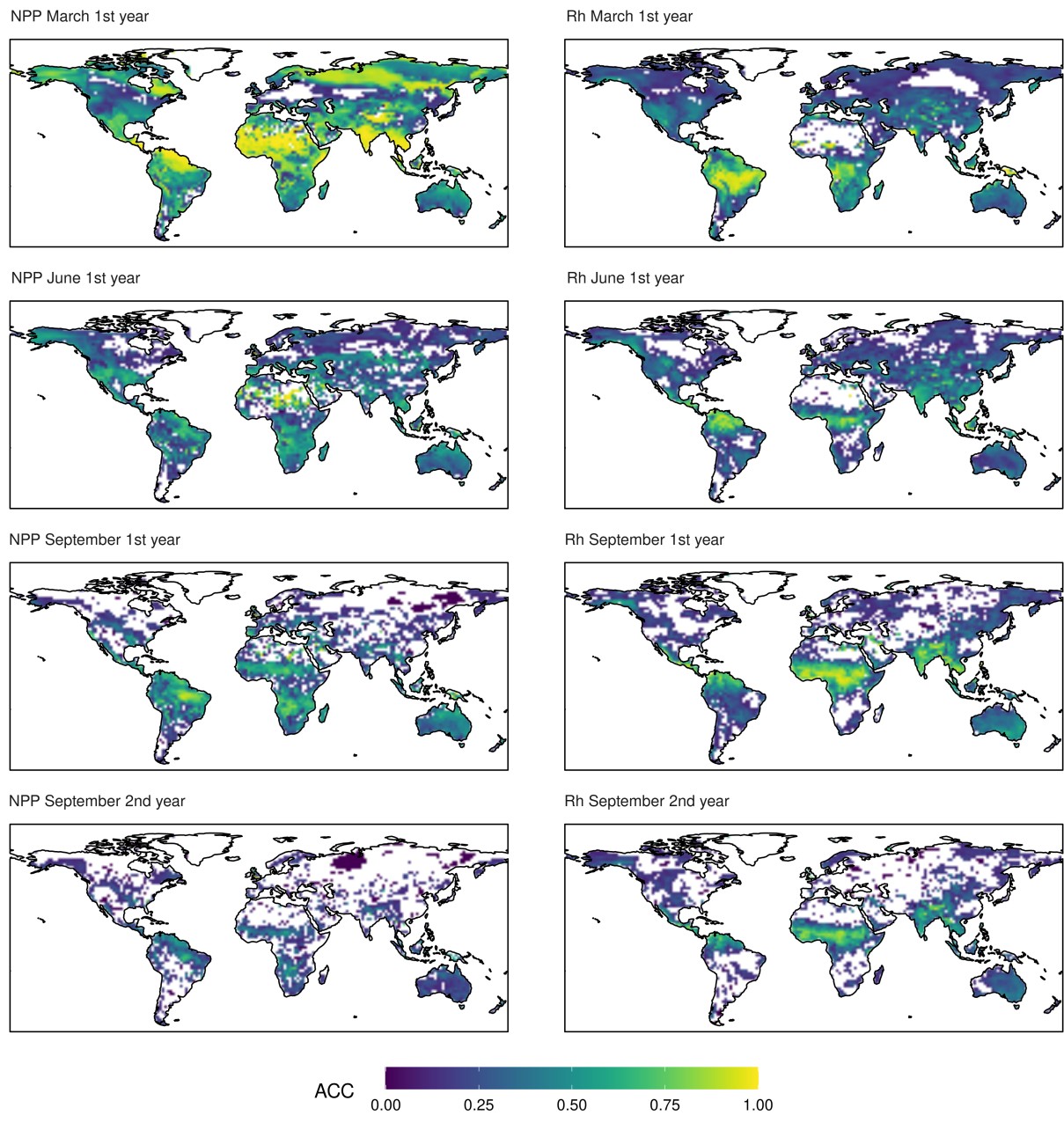

**Figure 3.** ACC of NPP and Rh. The color scale is cropped at zero. Only values above the 95% confidence interval are shown.

degradation due to overgrazing in dry years (Tews et al., 2006). Other promising regions are Northeast and central Brazil. The high NPP*pred* in these areas could be used to select of crop varieties which are more or less drought tolerant depending on the given forecast.

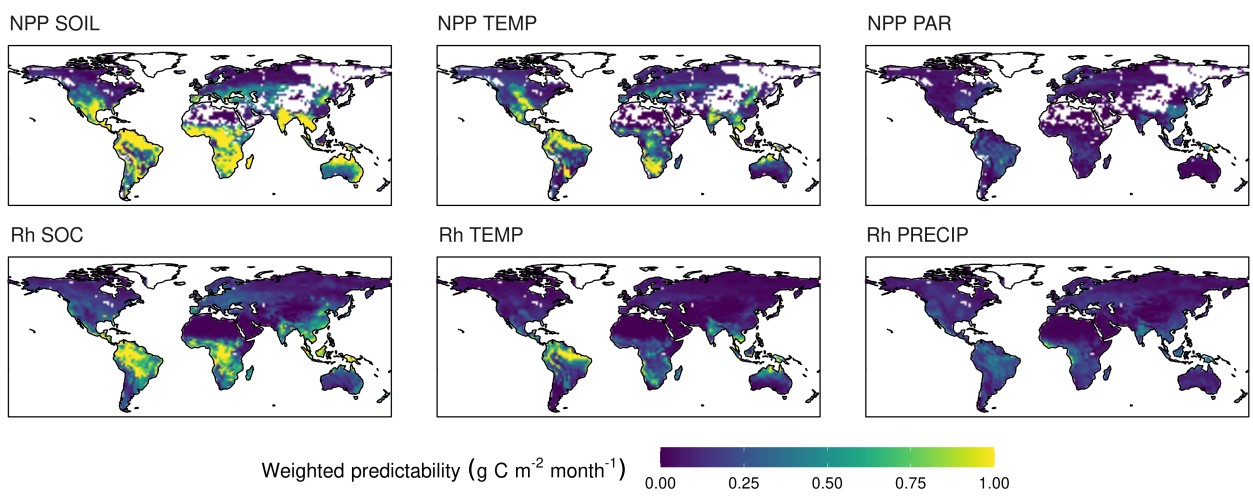

**Figure 4.** Contributing components to the weighted predictability ($wV_c$) of NPP and Rh. The contributors to NPP predictability are the predictability of soil moisture (SOIL), temperature (TEMP) and photosynthetically active radiation (PAR). Contributors to Rh predictability are the predictability of soil organic carbon (SOC), TEMP and precipitation (PRECIP). The averaged predictability of the first 12 months lead time weighted by carbon flux anomaly of the ensemble means. The sum of all components of a flux type give the modelled total predictability of that flux.

## 3.2  Composition of predictability

CF*pred* is sufficiently captured by the regression models (4) with a correlation of 0.71 and 0.75 for NPP and Rh, respectively (averaged correlation between the $V_c$ derived from the ensemble simulations and the $V_c$ of the regression model for each grid cell and lead time, not shown). The contributors of CF*pred* show strong spatiotemporal heterogeneity with drivers alternating across seasons and regions. The temporally averaged contributions to weighted predictability are shown in Figure 4. The drivers of NPP*pred* are SOIL*pred* (sum of midSOIL*pred* and deepSOIL*pred*) and TEMP*pred*, which explain 62 and 30% of

the globally averaged NPP*pred*, respectively. PAR*pred* only contributes 8% to the NPP*pred*, most of it in the first month of the simulations. The NPP*pred* patterns of $V_c$ explained by SOIL*pred* and TEMP*pred* are similar to the patterns of ACC, although areas with low carbon flux densities are excluded through weighting by absolute flux anomaly. While the NPP*pred* explained by SOIL*pred* has a spatial extent that broadly covers all regions of high NPP*pred*, TEMP*pred* is concentrated to certain areas. TEMP*pred* is high in a band spanning from the Amazon basin to northern South America, in southern Africa and in Southeast

Asia. The largest contributor to Rh*pred* is SOC*pred* (52%) followed by TEMP*pred* (27%). Similar to NPP*pred*, the temperature component is highest in the Amazon basin, southern Africa and Southeast Asia.

In order to facilitate a system for operational NPP prediction, a network of sensors could be installed to gather data on the initial condition of the land surface. The patterns of the role of soil moisture in predicting NPP (Fig. 4) reveal the areas on which the efforts in establishing such a network should be focused on to maximize the impact.

There are more variables that are regarded as key drivers of NPP variability and could have been considered as predictors in the regression models. Most importantly, LAI and humidity play an important role in NPP variability (Schaefer et al., 2002). Several studies show the role of a dynamical simulation of LAI in extending the predictability of land surface processes (Zeng et al., 1999; Wang et al., 2010, 2011; Weiss et al., 2012, 2014). Here, the inclusion of LAI as a predictor is rejected because of the susceptibility of regression models to correlated predictors. The changing concentration of atmospheric $CO_2$ is causing trends in NPP as global atmospheric levels are rising (Winkler et al., 2021), however, we assumed that the interannual variability of $CO_2$ fertilization is below a meaningful contribution to overall variability. Although clay content plays a major role on carbon turnover rates in soil (Coleman et al., 1997), it is not considered in the JSBACH Rh submodel (Tuomi et al., 2009) and was not included in this study.

### 3.2.1 Seasonality

The seasonal patterns of NPP*pred* revealed in the ACC data (Fig. 2 and 3) are reproducible by the decomposed predictability metric $V_c$ (Fig. 5). They show the reemergence of predictability in the dry season at various locations, and reveal that this phenomenon can not be contributed to a single factor. The largest pattern is a reemergence in July to November at 1 ° to 4 ° South and can be associated with the high NPP*pred* in the southern Amazon (Fig. 3 NPP September 1st year). This pattern is due to increased TEMP*pred* throughout the dry season, which is extended by high deepSOIL*pred* in September, and even reoccurs in the second year of the simulation. Another pattern explains the high NPP*pred* in southern Africa between August and October, which is due to deepSOIL*pred*.

These cases of high dry season NPP*pred* in the tropics are most likely due to the seasonally changing limitations of NPP. During the productive wet season, plant growth is limited by incoming radiation (Wang et al., 2010) which has little variability and poor predictability. Instead, most of the interannual variability of NPP can be explained by dry season variability. One study found over 80% of western Amazon NPP variability to take place between July and September (Wang et al., 2011). The water limitation of NPP during the dry season (Tian et al., 2000) does not only introduce higher variability as compared with the energy limited wet season, but the coupling of NPP to soil moisture also lends NPP the high predictability of soil moisture.

Although the seasonality of Rh*pred* shows a reversed tendency to NPP*pred* with higher predictability in the wet season, the mechanisms explaining the seasonality are similar. The seasonally varying Rh*pred* can be explained by the inherently different predictability of the seasonally dominant limiting factor of Rh (Fig. 6). During the dry season the limiting factor of Rh is precipitation, which has a generally low predictability. The absence of precipitation for several weeks will inhibit soil respiration completely. There is a sharp increase in Rh variability in the dry-wet transition because the onset of precipitation is difficult to predict. As precipitation increases, the moisture constraint is asymptotically lifted and approaches zero. At this point, Rh becomes limited by substrate availability, which has a much higher predictability than climatic variables. The high SOC*pred* is due to the persistence of SOC anomalies because of the low decomposition rates and the pause of decomposition

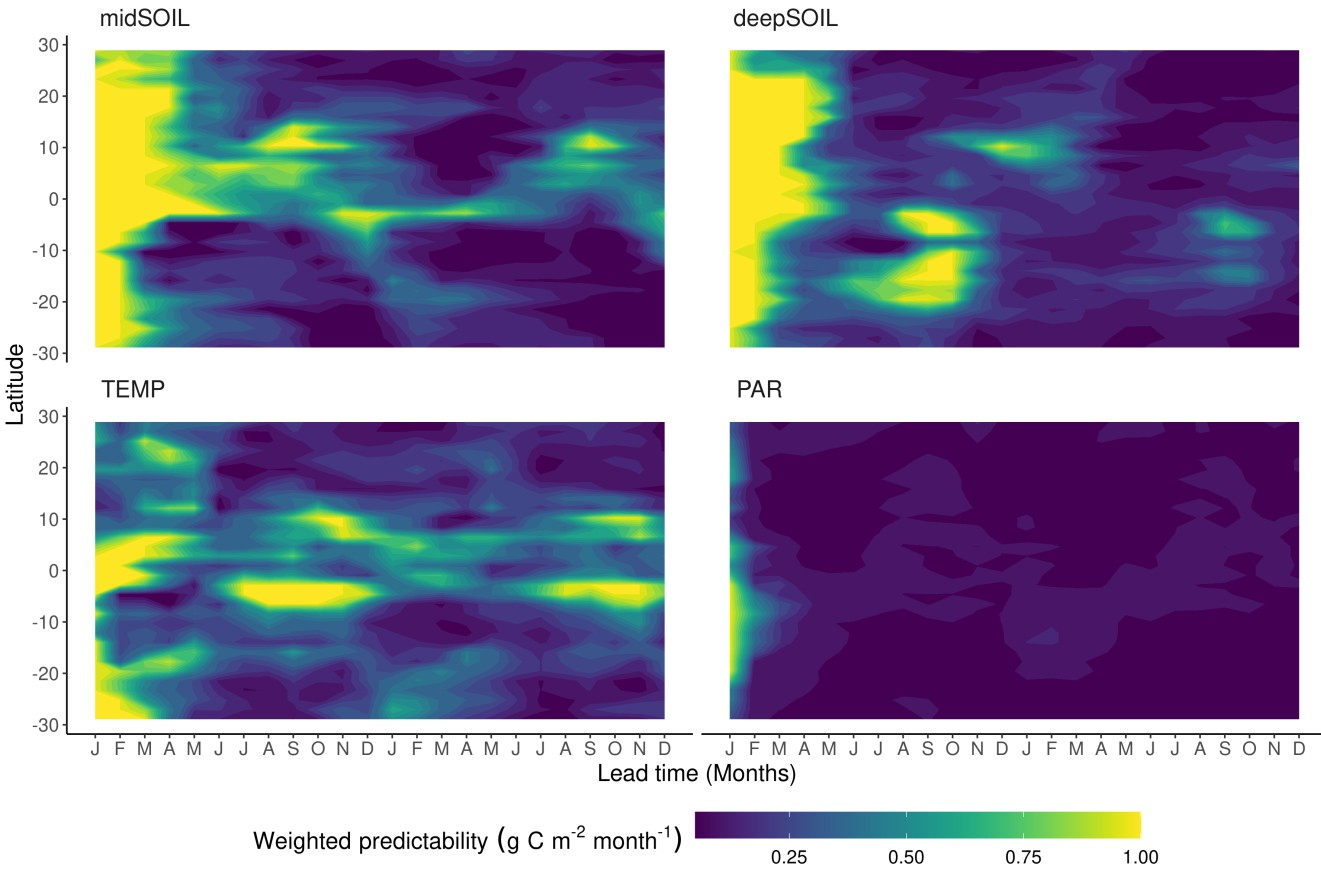

**Figure 5.** Zonal means of contributing components to the weighted NPP predictability ($wV_c$). The contributing components are the predictability of soil moisture in $19 - 78$ cm and $79 - 268$ cm depth (midSOIL and deepSOIL), air temperature (TEMP), and photosynthetically active radiation (PAR).

during dry seasons. Although TEMP*pred* is higher than PRECIP*pred*, it only plays a minor role in tropical Rh*pred*, because tropical Rh has relatively low temperature sensitivity (Meir et al., 2008).

These pronounced seasonal patterns of Rh*pred* hinge on the implementation of the precipitation sensibility function in MPI-ESM. The shape and parameterization of the rate modifying function of decomposition to moisture sets Rh to be more sensible to precipitation in the dry than in the wet season. However, the relationship between Rh and moisture in the tropics is the highly debated subject of various studies coming to different conclusions. These studies suggest a parabolic or no relationship with soil moisture (Meir et al., 2008) or linear increase with precipitation (Tian et al., 2000).

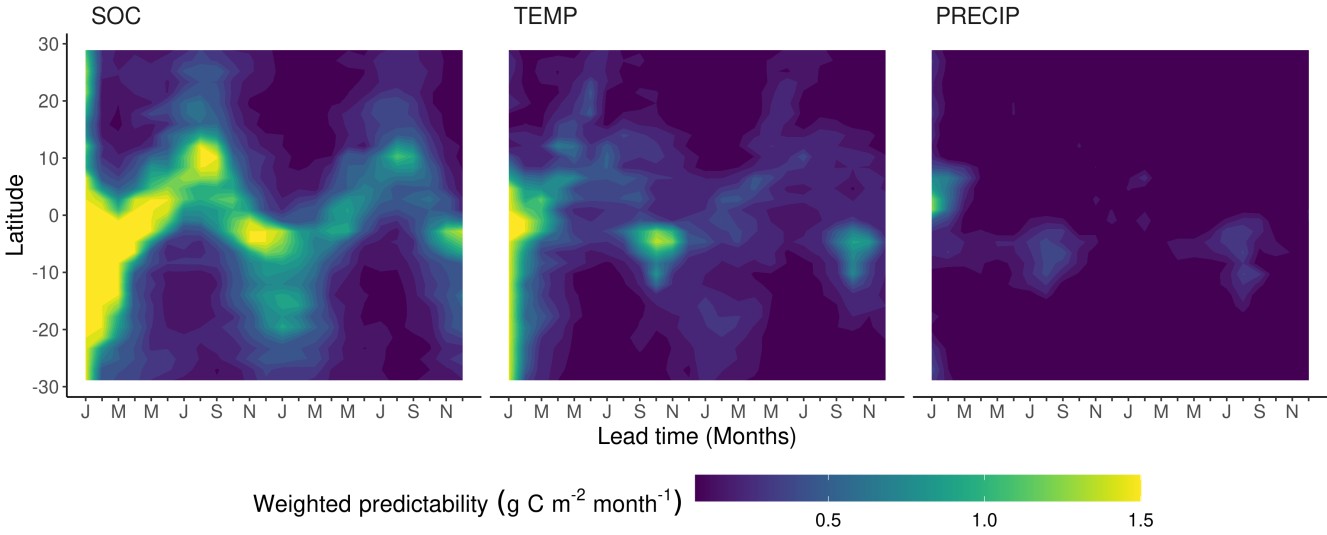

**Figure 6.** Zonal means of contributing components to the weighted Rh predictability ($wV_c$). The contributing components are the predictability of soil organic matter (SOC), air temperature (TEMP) and precipitation (PRECIP).

### 3.2.2 Interannual variability

Using the distance based predictability metric $V_c$ also allows to evaluate the variability of predictability between different initializations. Among the regions with the highest interannual variability of NPP*pred* are the southern Amazon basin (Box in Fig. 7), with a mean $V_c$ of 0.24 and an SD of 0.32, and northwestern Australia, with a mean of 0.16 and an SD of 0.60 (23 °S, 122 °W). Figure 7 shows how the interannual variability of NPP*pred* is affected by initial soil moisture. The majority of regions with a high NPP*pred* (Fig. 3 and 4) have a higher predictability in years initiated from wet states. Exceptions to this trend are India and northwestern Australia, where NPP*pred* is higher in dry years. The strongest difference in NPP*pred* is in the Amazon basin, where overall NPP*pred* and interannual variability of predictability are also at the global maximum.

To determine the mechanisms responsible for this difference in predictability we focus on the composition of the NPP*pred* in the southern Amazon basin (box in Fig. 7). To represent wet and dry years, a composite analysis is used based on the ENSO states (The El Niño years are the driest extremes at initialization while soils are often saturated at the beginning of La Niña years).

The different composition of NPP*pred* within the southern Amazon basin is shown in Fig. 8. La Niña years have an overall higher NPP*pred*, which even lasts throughout the second year of the simulations. However, the drivers causing the difference of increased La Niña predictability are changing over time. At the start of the growing season, which is between December and July, midSOIL*pred* contributes largely to the increased La Niña predictability while deepSOIL*pred* gains in importance around June, when topsoils begin to dry out. An increase in TEMP*pred* explains a large fraction of increased La Niña predictability throughout the first year.

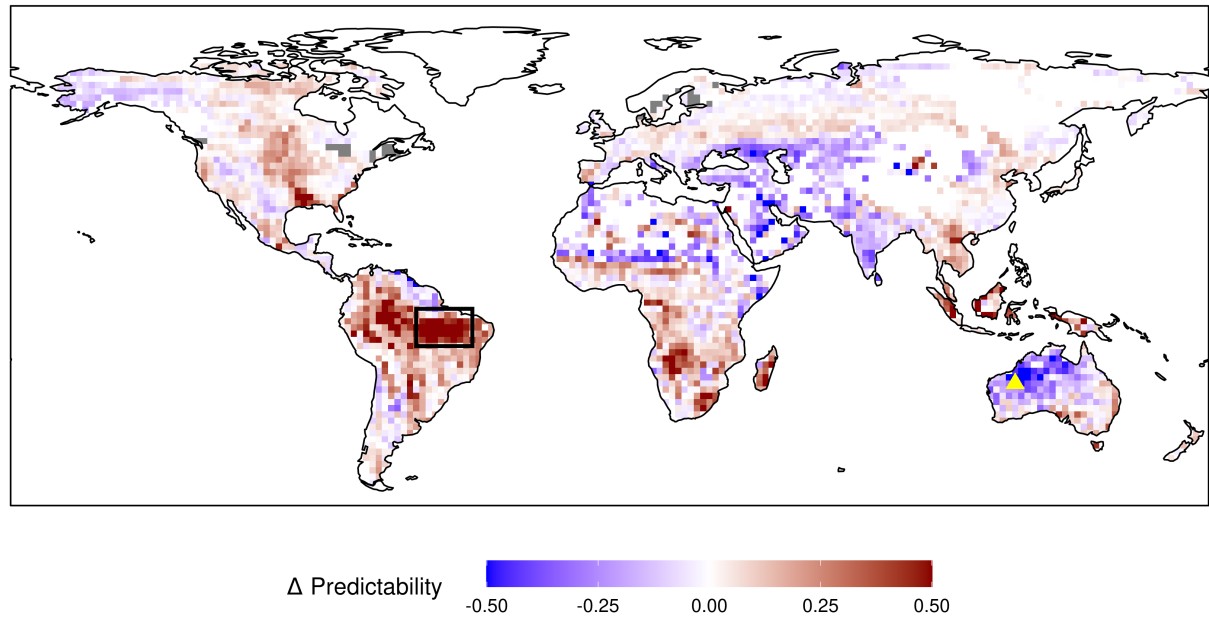

**Figure 7.** Difference in NPP predictability ($V_c$) based on the initial soil moisture. The mean NPP predictability of the first year from the 20% driest initializations are subtracted from the 20% wettest initializations for every grid cell. Red color means higher NPP predictability in wet years and blue color a higher predictability in dry years. Soil moisture from 19 – 78 cm depth is used to determine initial conditions. A large fraction of years included in the initializations are ENSO years, where the initial anomaly is further extended through persisting oceanic forcing. The black box and yellow triangle stand for regions examined in the main text.

The increase of midSOIL*pred* during the growing season can be explained by the relationship between precipitation and the change in soil moisture in spring (Fig. 9 (a)). Although the variability of precipitation is comparable between the ENSO states, there is little change in soil moisture in the La Niña years while the relationship between precipitation and soil moisture change is more pronounced in the El Niño years. The difference in this covariance between the ENSO states is linked to the initial water content (Fig. 9 (b)). The El Niño year is initialized at a depleted state, and precipitation is used to recharge midSOIL. This leads to the translation of the variability in precipitation to a variability in midSOIL. Since midSOIL is saturated at the initialization of the La Niña year, it is hardly affected by the variability of precipitation and the excess water leaves the system as runoff or drainage.

The same mechanism is responsible for the difference in deepSOIL*pred*. As midSOIL dries out during the summer months, NPP is increasingly coupled to deepSOIL. Every ensemble member of the La Niña simulation receives enough precipitation to saturate deepSOIL, thereby reducing its variability, while none of the members in the El Niño year can recharge the soil water deficit.

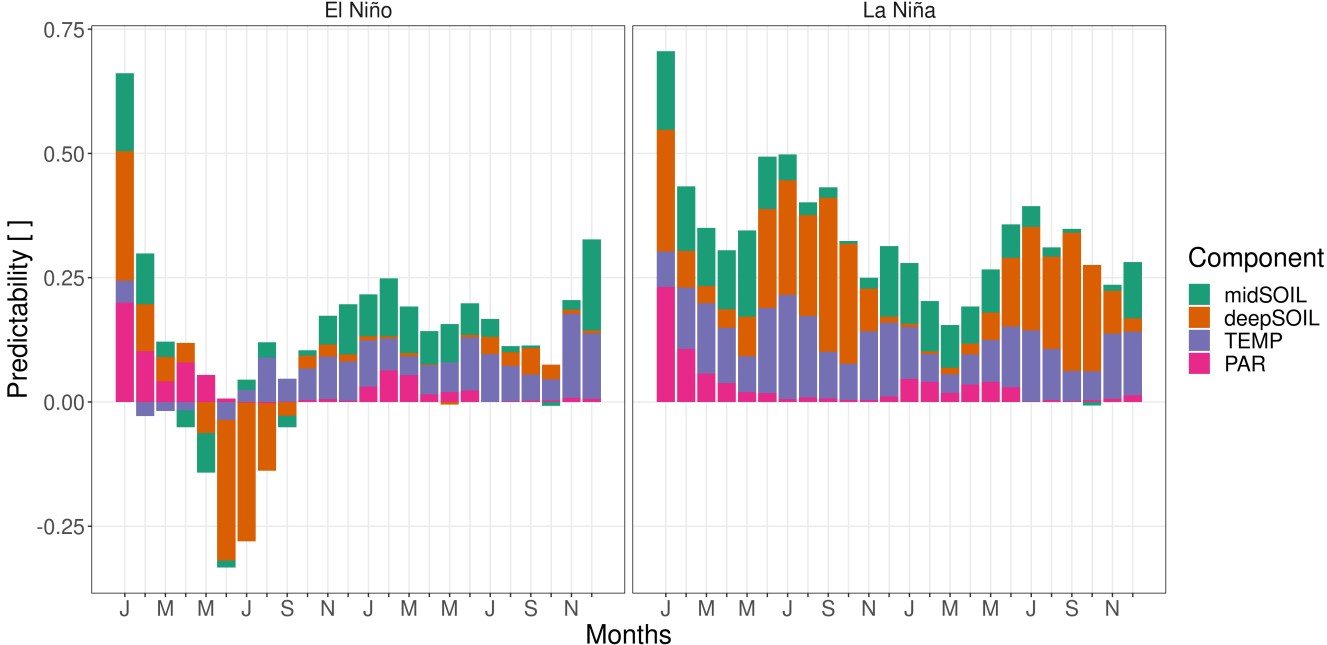

**Figure 8.** The composition of NPP predictability ($V_c$) in the Amazon basin by ENSO state. The contributing components are the predictability of soil moisture in 19 – 78 cm and 79 – 268 cm depth (midSOIL and deepSOIL), air temperature (TEMP), and photosynthetically active radiation (PAR). La Niña years have an overall higher predictability. Negative values mean an ensemble variance that is exceeding the climatological variance.

Increased NPP*pred* in wet years due to TEMP*pred* can have multiple reasons which are difficult to disentangle. As soil moisture and surface temperature are coupled through evapotranspiration, a reduced variability in soil moisture suggests a reduced variability in temperature as well. Contributing to this effect is the nonlinear mechanism controlling evaporation. At the wet end of the spectrum, evaporation is not limited by soil moisture, meaning that a small variability in soil moisture of a wet soil does not affect evaporation. A counteractive process that might increase predictability in dry years is described by Koster et al. (2011). They suggested that in ecosystems which are generally at the wet end of the spectrum (which is the case for the Amazon basin) land-atmosphere coupling is stronger in dry years when evaporation is limited by soil moisture. This increased coupling can extend TEMP*pred* by linking it to soil moisture. However, their study was conducted on North America, where land-atmosphere coupling is generally stronger than in tropical rainforest (Guo and Dirmeyer, 2013).

To investigate processes behind the difference in temperature variability per ENSO state, we analyzed the key elements of the surface energy balance. Almost all processes have a continuously higher variability in the El Niño years (Fig. 10). The strongest difference in variability is in net longwave radiation, but this is most likely an effect of increased variability of surface temperature and not the cause. The SD of net shortwave radiation and ground heat flux are evenly increased by around $0.4\,Wm^{-2}$ across the first year. Except for some winter and spring months, the latent and sensible heat fluxes have an increased

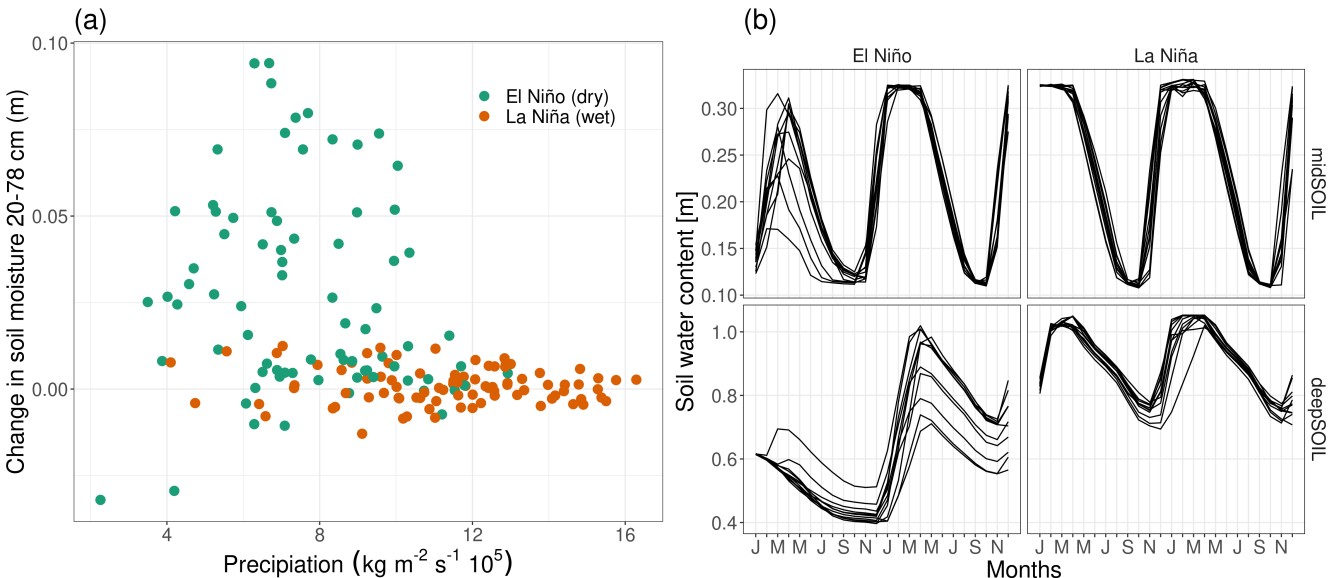

**Figure 9.** Soil water dynamics of different ENSO states in the Amazon basin at 8 °S, 54 °W. (a) Relationship between February precipitation and change in soil moisture from February to March. (b) Soil water content of the 11 member ensemble simulation for one specific El Niño and La Niña year (midSOIL and deepSOIL are the moisture content at 19 – 78 cm and 79 – 268 cm respectively).

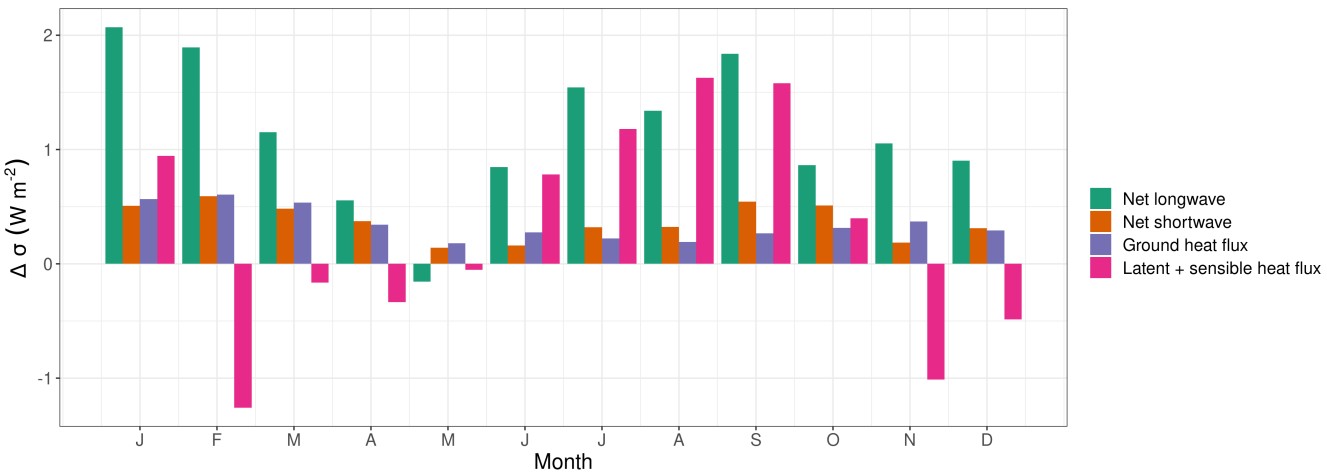

**Figure 10.** Difference in standard deviation ($\Delta\sigma = \sigma$ El Niño $- \sigma$ La Niña) of different components of the surface energy balance in the Amazon basin. The latent and sensible heat fluxes are pooled because of their strong negative correlation.

variability in the El Niño years. At the peak, difference in variability in August is mostly due to an increased variability in the latent heat flux.

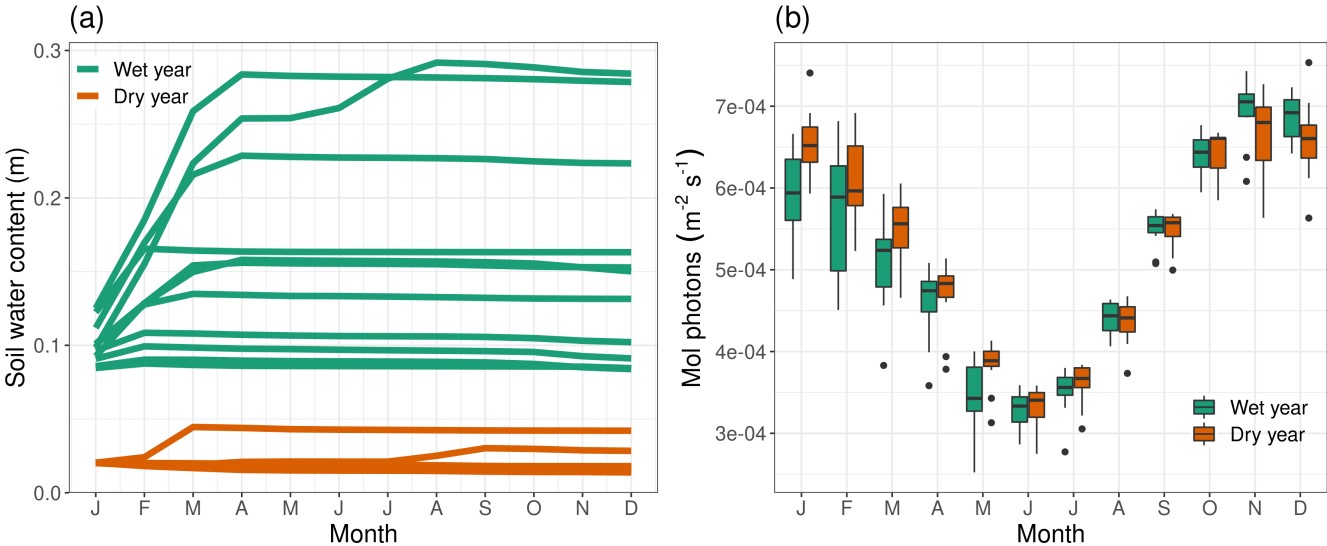

**Figure 11.** Difference in ensemble member spread in northwestern Australia between a wet and a dry year for (a) deepSOIL and (b) PAR.

As mentioned above, there are also certain regions with an inverse relationship between wetness and NPP*pred*. These are predominantly in arid regions like northwestern Australia, India, northern Caucasus and the western US (Fig 7). The mechanisms explaining the increased NPP*pred* in dry years are exemplified on two initialzations from the dry and wet spectrum in northwestern Australia at 23 ° South, 122 ° East (yellow triangle in Fig. 7).

This higher NPP*pred* can be contributed to less variability in deepSOIL and PAR (Fig. 11). The predictability providing mechanism of deepSOIL is comparable with the process in the Amazon basin. With soil moisture dynamics frequently operating at extreme ends of the water holding capacity, the variance can be minimized by all ensemble members being pushed against the boundaries of the system. As opposed to the Amazon basin, in northwestern Australia the ensemble members are are clustered at the dry end of the water holding capacity (Fig. 11 (a) dry years), while any introduction of soil moisture will increase the variability.

Another difference in NPP*pred* is caused by a differing variability of PAR (Fig. 11 (b)). Most dry years have little cloud cover and no restriction of incoming radiation. However, in wet years it is difficult to predict the extent of precipitation and cloud cover, which increases the variability of PAR.

The relationship between initial soil moisture and climate predictability is noted by others. Koster et al. (2011) have determined that, depending on the region, the direction of this relationship can go either way. This asymmetry of predictability is present in areas of high land-atmosphere coupling and is caused by the nonlinear relationship of evaporative fraction with soil moisture. Another study has investigated the predictability of European summer heat and find different weather regime frequencies in initially dry and wet conditions (Quesada et al., 2012). This study adds to the view that predictability is not a mere function of location, but depends on the state of the system and predictability therefore has a strong temporal variability.

## 4 Conclusions

In this study, we take a closer look at spatiotemporal patterns of terrestrial CF*pred*, identify the climatic and environmental sources of predictability, and the feedback mechanisms prolonging the memory of the system. We propose a metric of CF*pred* weighted by the amplitude of carbon flux anomalies. This metric allows to evaluate the role of different regions and processes to the predictability of the global carbon cycle.

We find that the spatiotemporal patterns of NPP*pred* and Rh*pred* are determined by (a) the predictability of the carbon flux drivers, (b) the climatic anomalies caused by low-frequency climate modes as ENSO, (c) the seasonal change of limiting factors and (d) threshold processes and nonlinearity of ecosystem responses.

On the global average, NPP*pred* is to 62% explained by SOIL*pred*, and to 30% by TEMP*pred*. Rh*pred* is explained by SOC*pred* and TEMP*pred* (50 and 27%) predictability. Decomposing the predictability signal shows there is a high spatiotemporal variability in the drivers of predictability. SOIL*pred* and SOC*pred* are distributed across all areas of high CF*pred*, while TEMP*pred* is mostly to be found in the Northern Amazon basin for CF*pred* and Southern Africa, North America and Southeast Asia for NPP*pred*. Rh*pred* can outlast NPP predictability because SOC, its main drivers, has a much higher anomaly persistence than the drivers of NPP. On the other hand, NPP is more directly affected by climatic drivers and is therefore able to benefit from the predictability of persisting climatic anomalies like the effects of ENSO. Intraannual variability of CF*pred* is controlled by the seasonally specific limiting factor of NPP and Rh. This leads to NPP gaining predictability in the dry season, when soil moisture replaces PAR as the limiting factor, while Rh*pred* has its peak in the wet season, when SOC drives the carbon fluxes instead of precipitation in the dry season. This change in limiting factors is due to the nonlinear relationships of transpiration to soil moisture and Rh to precipitation. Both of these relationships describe a saturation point, at which the variability of moisture (precipitation) becomes insignificant to carbon fluxes. Lastly, interannual variability of NPP*pred* reveals an asymmetry of predictability driven by initial soil moisture and subsequent precipitation. This effect is caused by ecosystems operating at the boundary conditions of the soil moisture regime. The ensemble members of predominantly wet ecosystems are harmonized in wet years when precipitation exceeds the water holding capacity and excess water is removed through runoff and drainage. The reversed effect applies for ecosystems operating at the dry end of the spectrum. These processes reduce the covariance between precipitation and NPP.

Our results highlight the sources of CF*pred* and can be used for model development to improve the representation of the terrestrial carbon cycle. Further research could be directed towards the simulation of the ENSO imprint in climate models and the relationship between soil moisture and terrestrial carbon fluxes.

*Acknowledgements.* The authors wish to thank Hongmei Li for her helpful and constructive comments. This paper contributes to the 4C project.

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
