# Peer review of "Process-based analysis of terrestrial carbon flux predictability"

_Earth System Dynamics, 2021_

## Referee Comment (RC2)

General comments:

This paper presents a study on terrestrial carbon flux predictability using MPI ESM, which shows NPPpred is driven by soil moisture predictability and Rhpred mainly by soil organic carbon (SOC) and explored the effects of climate variables and events (El Nino and La Nina) on Amazon and Australia areas' CFpred. The analyses contents are abundant and the results are convincing. The paper is well-written but needs further careful checks and polish on details.

Specific comments:

I have some questions on the manuscript.

1. The selected variables for NPP and Rh have good references and reasons, and it may need some explanations/discussions why not precipitation and CO2 [1] for NPP, and why not soil moisture, soil clay content [2, 3] (important for soil respiration) for Rh, and the different/related effects in precipitation and soil moisture for NPP and Rh (e.g. the time lag effect of soil moisture with precipitation).

2. Are there any conditions for the results of 62% for soil moisture for NPPpred and 52% for SOC for Rhpred (add words that this is for global mean, and discuss with key regions such as Amazon)? And it needs to be more specific for "reveal **the crucial regions and ecosystem processes** to be considered when initializing a carbon prediction system".

3. The scale mismatch problem between site observed data and model simulated results makes the comparison of NPP and Rh very difficult, and thus result the difficulties in reducing uncertainty in simulated terrestrial carbon fluxes. And this raises some questions on true meaning of calibrating models with site specific observations with several sets of parameters and their spatial representatives (line 30). Such mismatch may deserve discussions. And I cannot find the *o* (validation anomalies) descriptions for global gridded NPP and Rh. And some discussions of uncertainties in model structures such as the models involved in TRENDY may be needed.

Technical corrections and some minor comments:

1. Add "and" in line 10 between "soil organic carbon" and "temperature".

2. Extend implications of this study, for example, can the results here help to constrain the uncertainty in land sink projections?

3. Can add this ref Zeng et al., (2014) [4] in refine model structure (line 31-32);

4. Explain somewhat of "the perfect model framework" in line 36, and why is it called "perfect"?;

5. Why Fig.2, 5 and 6 only showed -30~30 instead of -90~90?;

6. What are "other factors" in Line 169; And why the Congo basin is not strongly affected by ENSO?

7. Fig.7 needs legend for black rectangle and yellow triangle and relevance with the following figures and analyses;

8. The long term effects of the initial soil moisture would become very weak for Fig.7? And blue color means lower NPP predictability in wet years in Fig.7 ?

9. Are there mechanisms in switch of deepSOIL and midSOIL for La Nina in Fig.8 from March to June?

10. Line 296, the driving factors can be different across key regions (such as discussions in Lines 169), can add some specific summary on key regions.

11. Line 413, delete space of "CO 2";

12. Lines 375-426, need to maintain reference formats such as to capitalize journal names (e.g. Functional plant biology; Global change biology; Global biogeochemical cycles).

References:

1. Wang, S., et al., *Recent global decline of CO2 fertilization effects on vegetation photosynthesis.* Science, 2020. **370**(6522): p. 1295-1300.

2. Coleman, K., et al., *Simulating trends in soil organic carbon in long-term experiments using RothC-26.3.* Geoderma, 1997. **81**(1-2): p. 29-44.

3. Wang, G., et al., *Modeling soil organic carbon dynamics and their driving factors in the main global cereal cropping systems.* Atmos. Chem. Phys., 2017. **17**(19): p. 11849-11859.

4. Zeng, N., et al., *Agricultural Green Revolution as a driver of increasing atmospheric $CO_2$ seasonal amplitude.* Nature, 2014. **515**(7527): p. 394-397.

---

## Author Response (AR1)

We thank the Editor and the reviewers for their constructive and helpful comments. Our point-by-point responses to comments are given below in red color.

Reviewer No.1:

1. The 'perfect model framework' is mentioned in the introduction long before it is explained in the methods section. Adding a paragraph to the introduction to explain what the method is, where it has been used before, and its limitations, would help frame the paper better.
The paragraph describing the perfect model approach was moved to the introduction. The term "upper limits of predictability" was added to stress the limitations of the method.

2. In the discussion section a paragraph could be added to explain the practical implications of the results. This is briefly mentioned in the last line of the conclusions but it could be fleshed-out better. As is, the paper does not do a good job of explaining why readers should care about the results.
The following sections were added to the discussion:
[On the role of ENSO]. These findings highlight the importance of correctly simulating the ESNO process. Especially the localization of ENSO related rainfall patterns is crucial, since they provide a sustained and predictable anomaly in water availability.

[On the regional patterns of high and low NPP predictability]. The results reveal different areas in which an operational NPP forecast can be used to increase food security. The high NPP*pred* of the Sahel and Kalahari savanna ecosystems (Fig. 3) could be used to plan stocking rates in order to avoid grassland degradation due to overgrazing in dry years (Tews et al. 2006). Other promising regions are Northeast and central Brazil. The high NPP*pred* in these areas could be used to select of crop varieties which are more or less drought tolerant depending on the given forecast.

[On the composition of drivers of predictability]. In order to facilitate a system for operational NPP prediction, a network of sensors could be installed to gather data on the initial condition of the land surface. The patterns of the role of soil moisture in predicting NPP (Fig. 4) reveal the areas on which the efforts in establishing such a network should be focused on to maximize the impact.

[On the high interannual variability of predictability]. This study adds to the view that predictability is not a mere function of location, but depends on the state of the system and predictability therefore has a strong temporal variability.

Line 21: Be clearer here about whether you mean the seasonal cycle or annual variation in the 1st derivative of CO2 concentration.
To our understanding interannual is the term used to describe the differences between years, while intra-annual would be used for variations within years (season to season).

Line 29: Unclear what "of emission reduction detection in the face of internal variability" means.

The section was changed to "...the difficulty to detect the efforts taken in emission reduction due to internal variability of atmospheric CO2 variability"

Line 39: Change 'here' to 'therein'
Done.

Line 68: An abbreviation for standard deviation seems unnecessary. Also was the abbreviation ever introduced?
The abbreviation is introduced in line 22. We find it to be a useful and common abbreviation.

Line 97: 'verification' is the wrong word to use here.
We changed it to 'validation'.

Figure 3: Rephrase caption to eliminate 'significant'. 'values above the 95\% confidence...' is good enough to convey the meaning with stepping on the land-mine of whether or not statistical significance is a metric that should exist.
Done.

Figure 7: Explain the yellow triangle in the figure caption.
Done.
* * *
Reviewer No. 2:

1. The selected variables for NPP and Rh have good references and reasons, and it may need some explanations/discussions why not precipitation and CO2 [1] for NPP, and why not soil moisture, soil clay content [2, 3] (important for soil respiration) for Rh, and the different/related effects in precipitation and soil moisture for NPP and Rh (e.g. the time lag effect of soil moisture with precipitation).

Both NPP and Rh are processes which are highly dependent on water availability. While the vegetation responsible for NPP can access multiple soil layers, most of Rh is taking place in the litter and the topmost soil layer. The moisture dynamics in these layers are more closely related to monthly precipitation then the moisture in the whole soil column (we will add a clarifying sentence on this in the manuscript). Additionally, the soil respiration sub-model is also using precipitation to calculate the rate of Rh. The following sentence was added to the methods: "Although precipitation has no direct relationship with Rh, the Rh submodel used in JSBACH is parameterized using precipitation, because of its strong relationship with moisture in the uppermost soil layer where most of the respiration takes place."
CO2 fertilization plays a large role in the prediction of the carbon flux trend. However we assumed that the interannual variability of near surface atmospheric CO2, which has an effect on seasonal to decadal

predictability, is low. We added this sentence to the discussion: "The changing concentration of atmospheric CO2 is causing trends in NPP as global atmospheric levels are rising (Winkler et al. 2021), however, we assumed that the interannual variability of CO2 fertilization is below a meaningful contribution to overall variability."

Clay content is not considered by the Rh model used in MPI-ESM (Tuomi et al, 2009). If the module would include clay content in the calculation of soil respiration, its effect could be measurable through the predictability of SOC, because clay content directly effects SOC dynamics. The following sentence was added: "Although clay content plays a major role on carbon turnover rates in soil (Coleman et al, 1997), it is not considered in the JSBACH Rh (Tuomi rt al, 2009) and was not included in this study."

Winkler, Alexander J., et al. "Slow-down of the greening trend in natural vegetation with further rise in atmospheric CO 2." *Biogeosciences Discussions* (2021): 1-36.
Tuomi, Mikko, et al. "Leaf litter decomposition—Estimates of global variability based on Yasso07 model." *Ecological Modelling* 220.23 (2009): 3362-3371.

2. Are there any conditions for the results of 62% for soil moisture for NPPpred and 52% for SOC for Rhpred (add words that this is for global mean, and discuss with key regions such as Amazon)? And it needs to be more specific for "reveal the crucial regions and ecosystem processes to be considered when initializing a carbon prediction system".
The stated numbers are now referred to as 'Global' results in the abstract. Further information on regional patterns was not added to the abstract due to text size limitations, but if the editor deems this information necessary we would include them.

3. The scale mismatch problem between site observed data and model simulated results makes the comparison of NPP and Rh very difficult, and thus result the difficulties in reducing uncertainty in simulated terrestrial carbon fluxes. And this raises some questions on true meaning of calibrating models with site specific observations with several sets of parameters and their spatial representatives (line 30). Such mismatch may deserve discussions. And I cannot find the o (validation anomalies) descriptions for global gridded NPP and Rh. And some discussions of uncertainties in model structures such as the models involved in TRENDY may be needed.
The reviewer is mentioning the difficulties arising from model parameterization based on site observations. We agree on the mentioned points, however, the validation of the carbon flux patterns is not within the interest of this study. The referred section in the text is included as an example for the efforts being made bring modelled processes closer to observations.

1. Add "and" in line 10 between "soil organic carbon" and "temperature".
Done.

2. Extend implications of this study, for example, can the results here help to constrain
See answers to comment 2 by reviewer number 1.

3. Can add this ref Zeng et al., (2014) [4] in refine model structure (line 31-32);

The publication is a good example of refined models to improve the simulation of the terrestrial carbon cycle and added it to the list.

4. Explain somewhat of "the perfect model framework" in line 36, and why is it called "perfect"?
Description is moved from the methods section to the introduction.

5. Why Fig.2, 5 and 6 only showed -30~30 instead of -90~90?;
The following was added to the results: 'Zonal plots of predictability are limited to 30° South to 30° North to highlight the areas of high predictability'

6. What are "other factors" in Line 169; And why the Congo basin is not strongly affected by ENSO?
The sentence was rephrased to: "It shows that the high NPP$pred$ of the tropics is not an intrinsic property of these ecosystems." - without referring to possible "other factors" determining the predictability. The low influence of ENSO on the Cogo basin is based on the findings of Holmgren et al, 2001.
Holmgren, Milena, et al. "El Niño effects on the dynamics of terrestrial ecosystems." *Trends in Ecology & Evolution* 16.2 (2001): 89-94.

7. Fig.7 needs legend for black rectangle and yellow triangle and relevance with the following figures and analyses;
The proper description was added to the figure caption.

8. The long term effects of the initial soil moisture would become very weak for Fig.7? And blue color means lower NPP predictability in wet years in Fig.7 ?
The figure caption was extended by this sentence: "A large fraction of years included in the initializations are ENSO years, where the initial anomaly is further extended through persisting oceanic forcing. The black box and yellow triangle stand for regions examined in the main text."

9. Are there mechanisms in switch of deepSOIL and midSOIL for La Nina in Fig.8 from March to June?
We recognize that the old formulation did not consider the shift in importance between midSOIL and deepSOIL during the growing season. The text was changed: "At the start of the growing season, which is between December and July, midSOIL$pred$ contributes largely to the increased La Niña predictability while deepSOIL$pred$ gains in importance around June, when topsoils begin to dry out."

10. Line 296, the driving factors can be different across key regions (such as discussions in Lines 169), can add some specific summary on key regions.
The following sentence was added to the mentioned part of the conclusion: "SOIL$pred$ and SOC$pred$ are distributed across all areas of high CF$pred$, while TEMP$pred$ is mostly to be found in the Northern Amazon basin for CF$pred$ and Southern Africa, North America and Southeast Asia for NPP$pred$."

11. Line 413, delete space of "CO 2";

Done.

12. Lines 375-426, need to maintain reference formats such as to capitalize journal names (e.g. Functional plant biology; Global change biology; Global biogeochemical cycles).
Done.